# *Calendula officinalis*—A Great Source of Plant Growth Promoting Endophytic Bacteria (PGPEB) and Biological Control Agents (BCA)

**DOI:** 10.3390/microorganisms11010206

**Published:** 2023-01-13

**Authors:** Polina C. Tsalgatidou, Eirini-Evangelia Thomloudi, Kallimachos Nifakos, Costas Delis, Anastasia Venieraki, Panagiotis Katinakis

**Affiliations:** 1Laboratory of General and Agricultural Microbiology, Agricultural University of Athens, Iera Odos 75, 11855 Athens, Greece; 2Department of Agriculture, University of the Peloponnese, 24100 Kalamata, Greece; 3Laboratory of Plant Pathology, Agricultural University of Athens, Iera Odos 75, 11855 Athens, Greece

**Keywords:** *Calendula officinalis*, plant growth promoting endophytic bacteria, biological control, *Botrytis cinerea*, detached fruit assay

## Abstract

The application of beneficial bacteria may present an alternative approach to chemical plant protection and fertilization products as they enhance growth and resistance to biotic and abiotic stresses. Plant growth-promoting bacteria are found in the rhizosphere, epiphytically or endophytically (Plant Growth Promoting Endophytic Bacteria, PGPEB). In the present study, 36 out of 119 isolated endophytic bacterial strains from roots, leaves and flowers of the pharmaceutical plant *Calendula officinalis* were further identified and classified into *Bacillus*, *Pseudomonas*, *Pantoea*, *Stenotrophomonas* and *Rhizobium* genera. Selected endophytes were evaluated depending on positive reaction to different plant growth promoting (PGP) traits, motility, survival rate and inhibition of phytopathogenic fungi in vitro and ex vivo (tomato fruit). Bacteria were further assessed for their plant growth effect on *Arabidopsis thaliana* seedlings and on seed bio-primed tomato plantlets, in vitro. Our results indicated that many bacterial endophytes increased seed germination, promoted plant growth and changed root structure by increasing lateral root density and length and root hair formation. The most promising antagonistic PGPEB strains (Cal.r.29, Cal.l.30, Cal.f.4, Cal.l.11, Cal.f.2.1, Cal.r.19 and Cal.r.11) are indicated as effective biological control agents (BCA) against *Botrytis cinerea* on detached tomato fruits. Results underlie the utility of beneficial endophytic bacteria for sustainable and efficient crop production and disease control.

## 1. Introduction

In the world’s history, natural products and especially plants have been extensively used by traditional medicine and pharmacy to promote human protection against many diseases [1]. Medicinal plants produce a wide range of biologically active compounds enhancing plant resistance in various biotic and abiotic stresses [2,3,4]. The composition and amount of the producing secondary metabolites depend on a number of factors such as plant species, plant age, environmental and soil conditions as well as plant association with microbes [5,6,7,8,9]. Simultaneously, the beneficial microbiome is affected by plant-producing secondary metabolites, which have an impact on the nature and physiological properties of their own producing bioactive compounds [10,11,12]. The induction of secondary metabolites from endophytic bacteria hosting aromatic and medicinal plants is widespread and is gaining a lot of interest because of their multidimensional properties [13,14,15]. Bacterial endophytes are beneficial microorganisms, directly associated with their host plant, living most of their life cycle inside plant tissues without causing any apparent symptoms of disease [16,17,18]. The bacterial endophytic microbiome has been distributed in all plant organs including seeds, roots, flowers, leaves and stems [19,20]. They produce various active metabolites, stimulate plant growth, increase nutrient acquisition and tolerance to abiotic stress factors, induce systemic resistance and protect host plants from various phytopathogens [21,22,23,24,25]. Due to their multiple plant growth-promoting (PGP) functions, endophytic bacteria can be characterized as plant growth-promoting endophytic bacteria (PGPEB). Plant stimulation leading to plant growth and root architecture development depends on the level of plant endogenous hormones and nutrient uptake, or it can be substantially influenced by PGPEB through direct expression of hormones and nutrient absorption like IAA production or phosphorus and iron solubilization [26,27,28]. Thus, isolation of cultivable endophytic bacterial strains with PGP traits has been demonstrated to stimulate plant growth and protection in field experiments and in vitro applications [29,30,31,32].

*Calendula officinalis* is a native medicinal plant with a range of pharmaceutical and medicinal abilities, hosting multiple endophytic and rhizospheric bacteria [33,34,35]. Growing under dry environments with high temperature, *C. officinalis* could establish a beneficial relationship with bacterial endophytes improving growth, yield and resistance of plant to different biotic and abiotic stresses. The objectives of the present study were (1) the isolation and characterization of endophytic bacteria associated with *C. officinalis* grown in the experiment field of Agricultural University of Athens in Spata, Greece; (2) screening the bacterial culturable endophytes for beneficial plant traits in vitro with a goal of detecting the most promising strains; (3) evaluation of their biocontrol activity in vitro and ex vivo; and (4) studying their ability to elicit plant growth promotion of *Arabidopsis thaliana* and *Solanum lycopersicum* var. Chondrokatsari Messinias in vitro.

## 2. Materials and Methods

### 2.1. Isolation of Endophytic Bacterial Strains from C. officinalis Plants

Endophytic bacteria used in this study were isolated from the pharmaceutical plant *C. officinalis* which was selected from the experimental field of Agricultural University of Athens in Spata in 2017. Fresh and healthy root, leaf and flower samples were washed thoroughly under running water and surface sterilized with 70% (*w/v*) Ethanol for 5 min, 0.5% (*w/v*) NaOCl- 0.2% (*w/v*) Tween20 for 5 min and 70% (*w/v*) Ethanol for 30 s, and were finally rinsed off four times with sterilized distilled water [36,37]. The efficiency of the sterilization method was determined by inoculating a sample of the last sterilized water into 1.5% (*w/v*) nutrient agar (NA) plates. The surface-sterilized material plant tissues were mashed in sterilized porcelain mortars and inoculated into NA plates. The plates were sealed with parafilm and incubated at 30° for 2–4 days in a bacteriological incubator until the appearance of morphologically different bacterial colonies. Single bacterial colonies were repeatedly streaked into fresh nutrient agar plates in order to obtain a purified endophytic bacterial isolation. Finally, the purified isolates were stored in −80 °C in 20% (*v/v*) glycerol stocks.

### 2.2. Molecular Characterization of Endophytic Bacteria Based on 16S rRNA

Endophytic bacterial isolates were grown in nutrient broth medium (NB) at 30 °C for 20 h. Genomic DNA was isolated withCTAB (Cetyl Trimethyl Ammonium Bromide) using the modified protocol by Moore et al. (1999) [38]. Universal primers FP: 5′-AGAGTTTGATCCTGGCTCAG-3′ and RP: 5′-AAGGAGGTGATCCAGCC-3′ were used to amplify 1500 bp region of 16S rRNA gene fragments and get sequenced [39]. The PCR mixture contained 1 μL of each primer, 5 μL 10× PCR Buffer, with Mg^2+^, 1 μL dNTPs, 0.5 μL Hot Start DNA polymerase, 1 μL DNA template and 40, 5 μL of autoclaved water in a final volume of 50 μL. DNA amplification was performed with the following thermocycler regime: 10 min at 94 °C (initial denaturation), followed by 32 cycles for 30 s at 94 °C (denaturation), 30 s at 48 °C (annealing), 1 min at 72 °C (extension) and a single step at 72 °C for 5 min (final extension). A total volume of 5 μL of each PCR product was loaded and visualized after electrophoresis in 1.2% (*w/v*) agarose gel. The PCR products were purified using GeneJet PCR Purification Kit from Thermo Fisher Scientific (US) following the instruction provided by the manufacturer. The purified partial 16S rRNA sequenced were deposited in GenBank with accession number as presented in Appendix A. The partial sequences of nucleotides were compared with available sequences from NCBI databases and sequences showing >99% similarity were retrieved by Nucleotide Basic Local Alignment Search Tool (BLAST N) program available at the National Center for Biotechnology Information (NCBI) BLAST server (www.ncbi.nlm.nih.gov/BLAST, accessed on 5 December 2022) [37]. A phylogenetic tree was constructed based on 16S rRNA gene sequence of the selected bacterial strains and type strains of NCBI database, under the Neighbor-Joining method in MEGA11 [40].

### 2.3. In Vitro Screening for Plant Growth Promoting (PGP) Traits

Bacterial endophytes were further categorized depending on their in vitro plant growth promoting (PGP) activities such as phosphate solubilization, siderophore, IAA, acetoin, lytic enzymes’ (chitinase, cellulase, protease) production and secretion of biosurfactants the liquid cell culture. A bacterial liquid overnight culture in NB was prepared for all the bacterial strains before inoculation in the specific culture media tested. For agar-based media, an artificial well was created in the center of the Petri dish plates with a 3 mm diameter cork borer. For phosphate solubilization ability, bacteria were spot inoculated (10 μL) in the well in Pikovskaya’s (PVK) agar medium (yeast extract 0.5 g, dextrose 10 g, calcium phosphate 5 g, ammonium sulphate 0.5 g, potassium chloride 0.2 g, magnesium chloride 0.1 g, manganous sulphate 0.0001 g, ferrous sulphate 0.0001 g, agar 15 g dissolved in 1000 mL distilled water) and assay plates were incubated for 2–5 days at 28 °C [41]. Phosphate solubilization activity was determined by the development of a clear halo around bacterial inoculation [42]. Siderophore production was determined using Chrome Azurol S (CAS) agar method by Schwyn and Neilands (1987) [43]. Exactly 10 μL of the bacterial liquid cultures were inoculated in the well of the CAS agar plates and incubated at 28 °C for 3–5 days. The development of a yellow-orange halo around the inoculation spot identified the siderophore production. For IAA production, endophytic bacteria were cultivated in NB supplemented with 0.1% (*w/v*) L-tryptophan for 48 h at 28 °C. Supernatant (2 mL) of the culture was obtained by centrifugation at 8.000 rpm/10 min and mixed with 2 drops of orthophosphoric acid and 4 mL of the Salkowski reagent (50 mL, 35% (*v/v*) of perchloric acid, 1 mL 0.5 M FeCl_3_ solution) [44]. Samples were incubated in the dark for 30 min at room temperature. The development of pink color confirmed the production of IAA. Optical density at 530 was measured to determine IAA production between bacterial isolates. For acetoin production the in vitro test Voges–Proskauer (VP-test) was conducted [45]. An aliquot of 10 μL of bacterial culture was inoculated in 5 mL of MR-VP broth and incubated at 30 °C for 48 h. Additionally, 2.5 mL of the culture was centrifuged for 15 min at 10.000 rpm and the supernatant was transferred in a glass tube. Then, after adding 600 μL of Barritt’s reagent A (5% (*w/v*) a-naphthol in absolute ethanol) and 200 μL of Barritt’s reagent B (40% (*w/v*) NaOH solution) to the broth, it was carefully shaken for 1 min and left to stand for 30 min after which the color development was recorded [45]. Chitinase activity was detected after inoculating 10 μL of bacterial culture into an artificial well on Nutrient Agar 1.5% (*w/v*) plates supplemented with 1% colloidal chitin [46]. After 72 h of incubation, the appearance of a clear halo around the well indicated chitinase production. Cellulase production was determined using CYEA (Casein-Yeast Extract Agar) (casein 5 g, yeast extract 2.5 g, glucose 1 g, agar 18 g dissolved in 1 L distilled water) medium supplemented with 1% (*w/v*) carboxymethyl cellulose [42]. Ten μL of bacterial overnight liquid culture was inoculated into the well and incubated at 28 °C. After 48–72 h, the agar plates were flooded with 0.1% (*w/v*) of Congo red solution for 15 to 20 min and then with 1 M NaCl for 15 to 20 min. The appearance of a pink-red halo around the inoculation spot indicated positive cellulase production [47]. Proteolytic activity was determined by spotting 10 μL of bacterial overnight culture onto CYEA plates supplemented with 7% (*w*/*v*) skim milk powder [48]. After a 72–96 h incubation at 28 °C, the formation of a clear zone around the well indicated positive proteolytic activity. Urease production was assayed by inoculating 10 μL of bacterial overnight culture into the artificial well on Christensen ISO 6579 urea base medium, ISO 19,250 (Conda, Madrid, Spain. After 72–96 h of incubation at 28 °C the formation of a pink halo around the well indicated ureolytic activity. Finally, a drop collapse assay was conducted for quick screening of biosurfactant production in the liquid cell culture as previously described in Tsalgatidou et al., 2022 [49].

### 2.4. Swarming, Swimming and Chemotactic Motility

Bacterial swarming, swimming and chemotactic motility were performed in vitro in Petri dishes. NA medium solidified with 0.5% (*w/v*) and 0.3% (*w/v*) agar was prepared to evaluate the ability swarming and swimming motility, respectively. A volume of 5 µL of a liquid bacterial overnight culture was inoculated in the center of the dish containing the semi-solid medium. Bacterial movement was captured (3 replicates/bacterial strain/treatment) after 24 h of incubation at 28 °C. Petri dish coverage was measured with ImageJ software v.1.8.0 [50] and was calculated with the formula (A1/A2) × 100, where A1: Bacterial cell coverage area and A2: plate area. To evaluate chemotactic motility, a vertical incision was made with a scalpel in the middle of a Petri dish containing NA with 1.5% agar, and half of the medium was removed with a spatula to be replaced with sterile potting soil with 50% of water content. A volume of 5 µL of a liquid bacterial overnight culture was inoculated in the soil part at 2 cm distance from the edge of the plate. Plates were placed vertically with the soil compartment being on the underside in order for bacteria to move from soil to NA and incubated for 48–36 h.

### 2.5. In Vitro Biocontrol Activity

Antimicrobial activity against plant pathogens *Rhizoctonia solani, Fusarium oxysporum* f.sp. *lycopersici* and *Botrytis cinerea* was estimated by dual culture assay. The pathogens were obtained from the Collection of Phytopathogenic Fungi from Benaki Phytopathological Institute, Kifissia, Athens, Greece and were stored on potato dextrose agar (PDA) at 4 °C until later use. A fungal disc of 5 mm diameter was placed onto NA (1.5% (*w/v*) agar) plates at a 3 cm distance from bacterial liquid cell culture (10 μL of 10^8^ CFU/mL) and bacterial cell-free culture (20 μL supernatant/well). After 7 d of incubation at 28 °C in darkness, biocontrol activity was evaluated depending on the mycelial inhibition either by contact or by forming a clear inhibition zone.

### 2.6. Bacterial Inoculation on A. thaliana Seedlings

Bacterial endophytes were firstly screened in vitro on *A. thaliana* Col-0 plantlets for elicitation of growth promotion. The *A. thaliana* Col-0 seeds were kindly provided by Dr Costas Delis (Department of Agriculture at the University of the Peloponnese). Seeds of *A. thaliana* were surface-sterilized with 5% NaOCl for 5 min, washed six times with sterilized distilled water and placed on Petri dishes containing half-strength Murashige and Skoog (½ MS) medium (Duchefa Biochemie, Haarlem, The Netherlands) supplemented with 0.5% (*w/v*) sucrose and solidified with 0.8% (*w/v*) bacteriological agar (Sigma, Burlington, MA, USA). After one day of stratification at 4 °C, the Petri dishes were transferred in a growth chamber under a long day photoperiod (16 h light with light/8 h of dark, with constant temperature set at 22 °C) [51]. Plates intended for bacterial inoculation on plant root tips or at distance from root tips were placed vertically in the growth chamber (6 plants/plate), unlike divided Petri dishes (I-plates) which were placed horizontally (4 plants/plate).

Endophytic bacterial strains were maintained on NA medium. A colony was inoculated in NB and grown overnight at 180 rpm/28 °C. Next-day aliquots of 10 μL of bacterial suspension (~10^8^ CFU/mL) were inoculated at the opposite site of the five-day-old seedlings at a distance of 3 cm from the root tip and plates were placed vertically in a growth chamber for ten more days. The root tips of seven-day-old seedlings were inoculated with 5 μL of bacterial suspension (~10^8^ CFU/mL) and plates were transferred in the growth chamber for eight more days. For the study of bacterial VOC’s on plant growth, after one day at 4 °C, 80 μL in total (4 × 20 μL) of bacterial culture (~10^8^ CFU/mL) was spotted in the opposite half of the I-plate, sealed with parafilm and placed horizontally in the growth chamber for three more weeks.

### 2.7. Phenotypic and Data Analysis of A. thaliana Plantlets

Seedling fresh weight of 12 plantlets of each inoculation treatment was determined immediately on an analytical balance. Digital images of Petri dishes of *A. thaliana* seedlings were captured using a digital camera positioned at the same distance from each sample. For root hair analysis, digital images were taken under an Olympus BX40 optical microscope and a section of 1 cm from the primary root tip was used for analysis. The ImageJ software v.1.8.0 was used to measure the primary root length and lateral root number of at least 12 seedlings and to quantify root hair number and length for bacterial treatments on root tip and at-distance from root tips. The effect of bacterial volatile production on plant growth was determined by measuring seedling fresh weight and leaf area using ImageJ software v.1.8.0 [50].

### 2.8. Bio-Priming of S. lycopersicum var. Chondrokatsari Messinias Seeds with Endophytic Bacteria

The effect of selected endophytic bacteria on germination and growth of tomato seedlings (*S. lycopersicum* var. Chondrokatsari Messinias) was studied under in vitro conditions as previously described in Thomloudi et al., 2021 [51]. The *S. lycopersicum* var. Chondrokatsari Messinias seeds were kindly provided by Dr Costas Delis (Department of Agriculture at the University of the Peloponnese). Selected strains were inoculated under sterilized conditions on tomato seeds. Germination rate was measured at 3 and 8 days post showing of bacteria-inoculated tomato seeds. When the experiment was completed, fresh weight, shoot length, primary root length and lateral root number were measured using ImageJ software v.1.8.0 [50].

### 2.9. Detached Fruit Assay

Selected bacterial strains were inoculated on small-sized detached tomato fruit (*S. lycopersicum* L. cv. Lobello) to evaluate their antagonistic capacity against *Botrytis cinerea* under ex vivo conditions as previously described by Tsalgatidou et al. (2022) [49]. Briefly, sterilized tomato fruit were artificially wounded and first inoculated with an aliquot of 10 μL of bacterial liquid culture (10^8^ CFU/mL) and incubated for 1 h prior to *B. cinerea* spore suspension injection (10 μL of 10^5^ spores/mL). Inoculated tomato fruit were transferred to plastic boxes, maintaining high humidity, and were incubated for 5 days in a growth chamber at 25 °C.

After five days of incubation, Disease Incidence (DI %) was calculated as the percentage of infected tomato fruits. The infected area of each tomato fruit by *B. cinerea* was measured using ImageJ software v.1.8.0 analysis tool and Disease Severity (DS %) was determined as the percentage of the infected area. Disease Severity Index (DSI %) was scored on a 0-to-9 rating scale, with 0 = healthy fruits, 1 = 1–10%, 3 = 11–25%, 5 = 26–50%, 7 = 51–75% and 9 = >75% infected fruit area and was calculated based on the formula: % Disease Severity Index, (DSI) = [∑(n × i)/(N × Z)] × 100, where n is the number of fruit in a specific value of disease rating scale, i is the corresponding value of the scale, N is the total number of fruit and Z is the highest value of disease rating scale.

### 2.10. Extraction and Evaluation of Bacterial Agar Diffusible Secreted Metabolites

Agar diffusible secreted metabolites of bacterial strains grown singly (B) or during confrontation with *B. cinerea* (B/F), were extracted as previously described by Tsalgatidou et al. (2022) [49]. Ethyl acetate extracts were further separated by thin-layer chromatography (TLC) onto silica gel 60 F254 TLC aluminum sheets with mobile phase consisting of chloroform–methanol–water at 65:25:4, *v/v/v*. Finally, separated bacterial extracts were evaluated for their antimicrobial activity against *B. cinerea* with TLC- bioautography assay as previously described by Tsalgatidou et al. (2022) [49] for strain Cal.l.30. The retention factor (R*f*) of each inhibition spot against the phytopathogen was calculated according to the formula R*f* = Distance travelled by the solute/Distance travelling by the solvent, to predict the possible antimicrobial compounds produced according to the literature.

### 2.11. Statistical Analysis

Data were statistically analyzed using IBM SPSS Statistics for Windows, version 25 (IBM Corp., Armonk, NY, USA). Statistical analysis was performed with ANOVA followed by Tukey’s honestly significant difference (HSD) test (*p* < 0.05) to allow for comparisons among all means and Dunnett’s test (*p* < 0.05) to compare bacterial treatments to the control values. Data expressed as percentages were arcsine-transformed prior to Tukey analysis (*p*-value < 0.05). Hierarchical cluster analysis was conducted in SPSS and heatmaps were created in Microsoft Excel 2010.

## 3. Results

### 3.1. Identification and Characterization of the Isolated Endophytic Bacterial Strains

A total of 119 cultivable endophytic bacterial strains were isolated and purified from leaves, roots and flowers of the pharmaceutical plant *C. officinalis* and were categorized based on their colonial morphology into 14 groups (A1-7, B1-4, C, D and E) (Figure 1). A representative number of 36 isolates were amplified and identified using the 16S rRNA gene sequence. The 16S rRNA gene Blast analysis revealed high similarity of the selected bacterial strains to five main genera including *Bacillus*, *Pseudomonas*, *Rhizobium*, *Stenotrophomonas* and *Pantoea* (Appendix A). Using the Neighbor-joining method, a phylogenetic tree was generated based on the partial 16S rRNA gene sequence of the selected endophytic bacterial strains and type strains from NCBI database (Figure 1). The phylogenetic analysis revealed that 23 of the identified isolates belong to the genus *Bacillus* closely related to *Bacillus velezensis* (A1), *Bacillus mycoides* (A2), *Bacillus subtilis* (A3), *Bacillus* proteolyticus group (A4), *Bacillus cereus* (A5), *Bacillus megaterium* (A6) and *Bacillus halotolerans* (A7) species. Ten isolates of group B1 to B4 corresponded to the genus *Pseudomonas*, formed subclades with *Pseudomonas frederiksbergensis*, *Pseudomonas kilonensis*, *Pseudomonas koreensis* and *Pseudomonas viridiflava*, respectively. Finally, Cal.r.35, Cal.r.8.2 and Cal.l.7a grouped amongst species of the genera *Rhizobium*, *Stenotrophomonas* and *Pantoea* and formed subclades with *Rhizobium nepotum*, *Stenotrophomonas rhizophila* and *Pantoea agglomerans*, respectively.

### 3.2. Plant Growth Promoting Activities

The selected bacterial endophytes were tested for their ability to solubilize phosphate, to produce siderophores, IAA, acetoin and different enzymes (cellulase, chitinase, urease and protease) and different strains were positive to different traits. The appearance of a clear (Figure 2A,D,F) or discolored (Figure 2B,C,E) halo around the point of inoculation in the solid growth media, as well as the discoloration of the liquid media (Figure 2K–L), indicated a positive result in each of the above assays.

From the data obtained, bacterial endophytes of *Pseudomonas* and *Bacillus* species cluster in distinct separate groups as presented in the Heatmap, with a positive reaction to different PGP traits (Figure 3). All the *Bacillus* strains tested produce cellulolytic and proteolytic enzymes, and the volatile compound acetoin. Almost half of them solubilize P, produce siderophores and urease, while 91.7% of them were positive for the production of the plant growth hormone IAA. In contrast, none of the *Pseudomonas* strains were able to produce acetoin in vitro, while all of them produced iron-chelating compounds and IAA, and solubilized P with the creation of a clear halo around their colony. Among all bacteria tested, only *Stenotrophomonas* sp. Cal.r.8.2 was positively detected for chitin hydrolysis in vitro.

The bacteria were also studied for their ability to move on different surfaces like swarming agar (Figure 2G), swimming agar (Figure 2H) and soil (chemotaxis) (Figure 2I). Furthermore, they were tested for their antagonistic activity against phytopathogenic fungi such as *Fusarium oxysporum* f.sp. *lycopersici* (Figure 2J), *Rhizoctonia solani* and *Botrytis cinerea.* Finally, all isolates were tested for their ability to secrete biosurfactant compounds in their liquid cell-free growth culture (supernatant) (Figure 2M) and their ability to reduce mycelial growth with the creation of an inhibition zone (Figure 4).

Thirty out of thirty-six isolates inhibited mycelial growth of both *R. solani, F. oxysporum* and *B. cinerea* either by contact (light blue) or by forming an inhibition zone (dark blue) (Figure 3). All the bacterial endophytes that inhibited the pathogens’ growth by forming an inhibition zone belong to the *Bacillus* species and more specifically to *B. subtilis* (Group A3), *B. halotolerans* (Group A7) and *B. velezensis* species (Group A1). Among all isolates, these bacterial strains spread better and covered the entire surface of the plate by swarming and swimming motility, and were the only strains positive for biosurfactant production in their supernatant (CFC), showing excellent drop collapse ability. The cell-free bacterial culture (CFC) was further tested for its ability to reduce mycelial growth of both *F. oxysporum*, *R. solani* and *B. cinerea* in vitro with the dual culture method (Figure 4). As presented in Figure 4, all bacterial strains forming a strong inhibition zone in a dual culture were able to suppress the pathogens with their cell-free supernatant, forming again a clear inhibition zone.

The rest of the *Bacillus* species suppressed the phytopathogens by contact, except *B. Mycoides* where no inhibition was observed. Finally, eight out of ten *Pseudomonas* strains that presented biocontrol activity inhibited all three fungi by contact (Figure 3). The bacterial strains presenting antagonistic activity by contact inhibition as liquid culture were not able to suppress the pathogens when their bacterial supernatant was inoculated against the pathogens.

### 3.3. Survival in Variable Growth Conditions

All selected bacterial strains were tested for their ability to survive in extreme growth conditions, such as high and low temperatures, extreme pH values and elevated salinity concentrations. The heatmap in Figure 5 presents the bacterial survival ability as well as the maximum cell concentration on a color scale (10^0^–10^8^ CFU/mL).

As presented in Figure 5, the optimum growth conditions of temperature and pH in which all strains reached their maximum concentrations are 25 °C and pH 6.8. At pH of 5.5 and growth temperature of 25 °C, the majority of the endophytes also grew to a high cell concentration. At pH 8.5, most bacterial strains of the *Pseudomonas* species reached a high cell density, as well as the individual strains of the *Stenotrophomonas*, *Pantoea* and *Rhizobium* species. On the contrary, at the highest incubation temperature of 45 °C, the majority of the isolates that managed to survive, reaching a cell concentration of 10^6^–10^8^ CFU/mL, belong to the *Bacillus* species. Among the 36 strains growing at 5 °C, strains Cal.r.20 and Cal.l.7a stood out reaching a cell density of 10^6^ and 10^7^ CFU/mL, respectively. Finally, inoculation in a growth medium with high salinity concentration (5%) resulted in a great reduction of bacterial cell density, not exceeding 10^3^ CFU/mL. Specifically, strains that manage to reach a final concentration of 10^5^–10^6^ CFU/mL are Cal.r.29, Cal.l.33, Cal.r.19, Cal.r.22, Cal.r.1, Cal.l.21, Cal.l.30 and Cal.f.2.1.

### 3.4. Effect on A. thaliana Growth Characteristics

All selected endophytic bacterial strains were co-cultured with *A. thaliana* (Col-0 ecotype) seedlings to evaluate their plant growth-promoting effect in vitro, after at-distance or on-root-tip formulation. Total fresh weight (FW), lateral root number (LRN), primary root length (PRL), root hair number (RHN) and root hair length (RHL) were measured to evaluate the effect of bacteria on a plant’s morphological and growth parameters. From the data obtained, each bacterial species was further categorized into one of four different groups based on *A. thaliana* root morphology after at-distance formulation from plant root tips (Figure 6). Therefore, the resulting categories comprise plants with i: long PRL and decreased LRN, ii: long PRL and increased LRN, iii: intermediate PRL and increased LRN and iv: short PRL and increased LRN (Figure 6).

Bacterial strains belonging to *B. mycoides* (Group A2), *B. proteolyticus* group (Group A4) and *B. cereus* (Group A5) species, as well as bacterial strain Cal.r.8.2 of the genus *Stenotrophomonas* (Group D), present plants with no remarkable morphological characteristics (morphological group i), similar to control plants (Table 1). They resulted in plants with low total fresh weight and as shown in Figure 7(Ai), plants with a decreased number of lateral root hair (Figure 7(Aii)) and root hair length (Figure 7(Aiii)).

In contrast, plants inoculated with bacterial strains from the species *B. velezensis* (Group A1), *B. subtilis* (Group A3) and *B. halotolerans* (Group A7) showed the best developmental characteristics, as described for morphological group ii (Table 1). Their positive effect was also observed microscopically, where representative strains Cal.r.29, Cal.r.19 and Cal.l.30 from bacterial groups A1, A2 and A7, respectively, presented plants with increased root hair number and root hair length (Figure 7(Aii)).

Finally, bacterial strains from *B. megaterium* species (Group A6) and all isolates from the *Pseudomonas* (Groups B1, B2, B3 and B4) genus presented plants from the morphological group iii, while bacteria *Rhizobium* sp. Cal.r.35 (Group C) and *Pantoea* sp. Cal.l.7a (Group E) present the plant’s morphological characteristics as described for group iv (Table 1). Representative bacterial strains Cal.r.20, Cal.r.21, Cal.l.6, Cal.r.6, Cal.r.35 and Cal.l.7a of each bacterial category, increased root hair number and root hair length of *A. thaliana* Col-0 seedlings/plantlets after at-distance formulation (Figure 7(Aii, Aiii)).

Bacteria-producing volatile compounds were also evaluated for their positive effect on *A. thaliana* plant growth. Bacterial strains Cal.r.29, Cal.r.19, Cal.r.33 and Cal.l.30 from the species *B. velezensis* (A1), *B. subtilis* (A3), *B. megaterium* (A6) and *B. halotolerans* (A7), respectively, resulted in plants with both increased total fresh weight and leaf area (Figure 7B). Plants inoculated with bacterial strains Cal.r.6, Cal.l.6 and Cal.l.7a from groups B3, B4 and E, respectively, presented plants with greater total fresh weight and no significantly increased leaf area, compared to control plants. On the contrary, bacterial strains Cal.r.20, Cal.r.21 and Cal.r.8.2 from groups B1, B2 and D, respectively, presented exactly the opposite results (Figure 7B).

### 3.5. Plant Growth Effect on S. lycopersicum var. Chondrokatsari Messinias Seedlings

The thirteen selected endophytic bacterial strains (Cal.r.29, Cal.l.30, Cal.f.4, Cal.f.5, Cal.r.11, Cal.f.2.1, Cal.l.11, Cal.r.33, Cal.r.19, Cal.l.21, Cal.r.20, Cal.l.7a and Cal.r.6) presenting the most significant plant growth promoting effect on *A. thaliana* seedlings and strong biological control potential were further inoculated on tomato seeds (*S. lycopersicum* var. Chondrokatsari Messinias) via seed bio-priming. In Table 2, the effect of bacteria on seed germination on the third (3 dps) and eighth day (8 dps) after showing of bacterial inoculated seeds.

Bacterial strain Cal.l.30 presents the highest percentage of germinated seeds on both days of observation, in comparison to strain Cal.l.21 that significantly reduced the seed’s germination. *Bacillus* strains Cal.f.4, Cal.l.11, Cal.r.19 and Cal.r.33 also increased significantly the tomato seed germination at the first time point (3 days post sowing, dps), while the differences were minimized during the eighth day of observation.

As shown in Figure 8A, the bacterial strains affected differently plant’s morphological characteristics such as total fresh weight, shoot and primary root length and plant’s lateral root number. In particular, bacterial strains Cal.r.29, Cal.l.30, Cal.f.4, Cal.f.5, Cal.f.2.1, Cal.l.11 and Cal.r.33 from the *Bacillus* species significantly increased plant fresh weight (Figure 8C) and total shoot length (Figure 8D) compared to control plants, while strains Cal.l.30, Cal.f.4, Cal.f.2.1 and Cal.l.11 were the most effective.

Strains Cal.l.21 and Cal.r.20 negatively affected all tomato plants’ characteristics, presenting the lowest total fresh weight, the shortest shoot and primary root length and decreased lateral root number. As shown in Figure 8E, tomato’s primary root length is not significantly influenced by most of the endophytic strains studied, except primary root lengths of plants inoculated with strains Cal.r.29, Cal.r.19, Cal.l.21, Cal.r.20 and Cal.l.7a that were shorter.

As presented in Figure 8F, plants’ lateral root number was significantly increased after Cal.l.30, Cal.f.4 and Cal.l.11 application compared to other applications or non- inoculated plants, with strains Cal.l.21 and Cal.r.20 effecting negatively this growth characteristic. Finally, all strains applied through seed bio-priming were able to colonize as single cells or aggregates of the tomato seedlings’ roots, showing a cell average of 10^3^ to 10^4^ CFU/seedling (Figure 8B).

### 3.6. Ex Vivo Biocontrol of Botrytis cinerea on Tomato Detached Fruit

The most promising bacterial endophytes combining plant growth promoting and antagonistic activity (Cal.r.29, Cal.l.30, Cal.l.11, Cal.f.4, Cal.r.11., Cal.f.2.1 and Cal.r.19) were tested against the fungal pathogen *B. cinerea* on detached tomato fruit. From the ex vivo application of both competitive endophytic bacterial strains and the phytopathogen, a significant reduction of gray mold disease was observed, compared to control fruits (*B. cinerea* application only). All bacterial strains successfully colonized the fruits after a day of inoculation, creating a visible, protective biofilm, and significantly reduced both the number of infected tomato fruit and the infected fruit area (Figure 9A).

The highest value of disease severity index, DSI (>72%), and disease incidence, DI (>91%) was observed for fruit treated only with *B. cinerea* (control inoculum). Treatments with endophytic strains Cal.f.4, Cal.r.29 and Cal.l.30 presented the healthiest tomato fruit with a significant reduction of DI at 13.33%, 15.0% and 25.0%, respectively and DSI at 12.85%, 11.49% and 17.49%, respectively (Table 3). Strains Cal.r.11, Cal.l.11, Cal.f.2.1 and Cal.r.19 followed with a significant suppression of *B. cinerea* as well. As presented in Table 3, all four strains reduced infected tomato fruits with disease incidence and disease severity index being under 39% and 31%, respectively.

### 3.7. Secretion of Bioactive Bacterial Agar Diffusible Secondary Metabolites When Grown Singly or against B. cinerea

Selected antagonistic endophytes from the *B. halotolerans* (Cal.l.30, Cal.f.4, Cal.r.11, Cal.l.11, Cal.f.2.1), *B. subtilis* (Cal.r.19) and *B. velezensis* (Cal.r.29) species created a strong inhibition zone during confrontation with *B. cinerea* in vitro. Agar-diffusible secreted metabolites produced by the endophytes when grown singly (B) or during interaction with *B. cinerea* (B/F) were evaluated for their antifungal activity using TLC-bioautography assay. Both B and B/F extracts of each bacterial strain suppressed *B. cinerea* spore formation and mycelial growth, forming strong inhibition spots with the same R*f* values (Figure 9B). All *B. halotolerans* species created two distinct inhibition spots similar between each strain with R*f* values of 0.36–0.37 and 0.39–0.40 (Figure 9B), similar to Cal.l.30, as previously described in Tsalgatidou et al., 2022 [49]. *B. velezensis* and *B. subtilis* extracts formed also two distinct inhibition spots with R*f* values of 0.35–0.36 and 0.39–0.40 for strain Cal.r.29 and of 0.48 and 0.55 for strain Cal.r.19 (Figure 9B).

## 4. Discussion

Medicinal plants are a great source of bacterial endophytes to study the relationship between the endophytes’ biodiversity in combination with their mechanisms of action and how they affect the host plant [12,28,52]. *C. officinalis* is one of the most important medicinal plants applied in traditional medicine from 12 BC until today, mainly due to its multitude of antimicrobial properties [53]. The endophytic bacterial community may directly contribute to the plant’s natural antimicrobial activity and positively affect plant growth, therefore, marigold is an excellent source of beneficial bacterial endophytes and strong BCA candidates [54]. To our knowledge, this the first report where endophytic bacteria from *C. officinalis* grown in Spata, Greece have been analyzed for their impact on plant growth promotion and biocontrol functions.

Using a cultivation-based approach, we successfully isolated and identified 36 fast-growing cultivable bacteria that were associated with leaves, roots and flowers of *C. officinalis*. The identified bacterial endophytes belong to *Bacillus*, *Pseudomonas*, *Pantoea*, *Stenotrophomonas* and *Rhizobium* species, which are the most common and frequently occurring species of endophytic bacteria [55]. A large number of bacteria belonging to the above genera have been extensively studied for their ability to enhance plant growth and protection, through a large number of direct and indirect mechanisms [56,57]. It is noteworthy that *B. halotolerans* was the predominant species among the *Bacillus* strains isolated, highlighting a possible ecological role in the vegetative stage of *C. officinalis*. The dominance of *Bacillus* genera has already been reported in other medicinal plants [58,59,60].

The endophytes of this study exhibited positive reaction to different beneficial PGP traits tested in vitro, such as secretion of iron-chelating compounds, production of the plant the hormone IAA, the volatile compound acetoin and various lytic enzymes (cellulase, chitinase, protease, urease), solubilization of phosphate, survival in different growth conditions, movement on different surfaces and production of antimicrobial compounds with strong biological control capacity, suggesting their multidimensional potentials for plant growth promotion and protection. Isolates belonging to the *Pseudomonas* species showed a high incidence of P-solubilization, IAA and siderophore production and a moderate incidence of cellulose and protease production, while none of the isolates produce urease or acetoin. In contrast isolates belonging to the *Bacillus* species had a high incidence in acetoin, urease, cellulose and protease production and moderate in P-solubilization, IAA and siderophore production. To support the biochemical assay results, all bacterial strains were first screened for short-term PGP effects on *Arabidopsis* seedlings under standard environmental condition on a Petri dish assay.

The screening involved enforced colonization of *Arabidopsis* seedling roots by applying each bacterial stain on the root tip, at 3 cm distance from the root tip and under complete separation from the plants, so interaction between plants and bacteria was mediated by emitted bacterial volatiles. The majority of the isolates, and specifically those from the *B. subtilis*, *B. velezensis*, *B. halotolerans*, *B. megaterium*, *Pseudomonas* spp., *Rhizobium* sp. and *Pantoea* sp. species, triggered plant growth parameters of *A. thaliana* seedlings after in vitro formulation through diffusible and/or volatile compounds, increasing total fresh weight and leaf area and through changing root architecture. Plant stimulation resulting to plant growth and modifications of root morphology depends on the level of endogenous plant hormones, or may be substantially be affected by PGPB through direct hormone expression and nutrient uptake [27,61]. High production of IAA can lead to plants with short primary root length, increased number and length of lateral roots and root hairs, allowing plants to increase the absorption of essential nutrients [26]. Bacterial endophytes from the above species reported the highest IAA production in comparison to low- or non-producing *B. mycoides*, *B. proteolyticus* and *Stenotrophomonas* sp. species, which did not significantly enhance plant growth and root development.

The exposure of plants to both IAA and bacterial volatile compounds (BVCs), such as acetoin or 2,3-butanediol, affects plant growth parameters and improves the root system, either directly or by regulating the level of IAA produced [62,63,64,65]. Therefore, different concentration of produced IAA in relation to the produced BVCs, probably affected the morphological characteristics of the root accordingly [66]. Specifically, bacterial strains Cal.r.35 (*Rhizobium* sp.) and Cal.l.7a (*Pantoea* sp.), combining the production of both acetoin and the highest level of indole-related compounds, resulted in plants with the shortest primary root length and the largest number of lateral roots and root hairs. The opposite results were observed when *Stenotrophomonas* sp. Cal.r.8.2, a negative IAA and acetoin producer, was inoculated in *Arabidopsis* plantlets. Furthermore, none of the *Pseudomonas* strains were positive for acetoin production, despite their plant growth stimulation when inoculated in divided Petri dishes, assuming they produce different active VOCs, not detected at present study. According to the literature, bacteria of the *Pseudomonas* species produce a variety of different VOCs contributing to plant growth stimulation and/or antifungal activity, such as 13-Tetradecadien-1-ol, 2-butanone and 2-Methyl-n-1-tridecene produced by *Ps. fluorescens* SS101 or methanethiol, dimethyl sulfide, dimethyl trisulfide, dimethyl disulfide (DMDS) and dimethylhexadecylamine (DMHDA) produced by *P. fluorescens* UM270 [67,68].

Plant growth stimulation was further observed after inoculating thirteen selected bacterial strains on tomato plants via seed bio-priming in vitro. Most of the bacterial strains tested increased and synchronized germination rate of tomato seeds compared to control plants, showing an outstanding effect on growth characteristics (e.g., total fresh weight, root hair number and root hair length, leaf area). In vitro-primed tomato seeds also led to a consistent positive performance of *B. halototolerans* strains (Cal.l.30, Cal.f4, Cal.l.11, Cal.r.11., Cal.f.2.1), *B. subtilis* Cal.r19 and *B. velezensis* Cal.r29, compared to the other *Bacillus* species strains as evidence by the significant increase in germination percentage, seedlings’ total fresh weight, shoot length and lateral roots, indicating the superior plant growth function of these bacterial strains. In accordance with the literature, similar results were observed from a number of experiments conducted under laboratory growth conditions after bacterial formulation on different plants [69,70]. Several bacterial strains from different species such as *Bacillus* spp., *Pseudomonas* spp., *Enterobacter* spp., *Azotobacter* spp. and *Burkholderia* spp. improved seed germination, stress tolerance and nutrient uptake after seed bio-priming formulation [71].

Seed priming with PGPB provides plants multiple benefits by improving physiological function and quality of seeds and enhancing plant growth, stress tolerance, disease resistance and protection [71]. PGPBs adhere to the seeds by forming a strong biofilm, hydrating the seeds thus activates and synchronizes their germination metabolic processes, while their multiple PGP traits affect the developmental features of the plants and yield [72,73]. Although previous studies have reported that various species of endophytic *Bacillus* strains isolated from medicinal plant have the potential to enhance root and shoot biomass and stimulate seed germination [74], this is the first report highlighting the PGP function of endophytic *B. halotolerans* species isolated from the medicinal plant *C. officinalis*. Recent reports also showed that endophytic *B. halotolerans* strains isolated from various plant species also exhibited an outstanding PGP activity under in vitro, green house and field conditions [51,75].

In order for a bacterial inoculant to be successful, it must effectively colonize the plant tissue it is applied on. PGPB develop a mutually beneficial relationship with the plant they colonize. Bacteria absorb essential organic compounds secreted from plant roots favoring their growth and colonization, simultaneously increasing the plants’ biomass and functional activity in the rhizosphere, inducing ISR mechanisms and protecting directly plants against various phytopathogens [76,77,78]. The ability of bacteria to colonize different plant tissues depends on several factors including their ability to move (e.g., swarming motility) towards nutrient-rich environments (chemotaxis), produce surfactants, form strong biofilm and eventually adhere to the application plant-surface [78,79,80]. Previous studies have shown that endophytes producing extracellular hydrolytic enzymes are involved in the indirect promotion of plant growth, as the secretion of these enzymes, namely cellulase and pectinase, could help them invade the cell wall, ultimately aiding in the colonization process of their roots [81,82,83]. In the present study, the selected *B. halototolerans* strains (Cal.l.30, Cal.f4, Cal.l.11, Cal.r.11., Cal.f.2.1), *B. subtilis* Cal.r19 and *B. velezensis* Cal.r29 performed the most extensive swarming, swimming and chemotactic motility in vitro and successfully colonized tomato seedlings’ root surface and the detached tomato fruits by creating a visible biofilm in the wound, thus protecting the fruits from the post-harvest pathogen *B. cinerea*.

A determinant factor involved in motility, chemotaxis and colonization ability for bacteria of the *Bacillus* species is the secretion of surface-active cyclic lipopeptides and specifically of surfactin [75,76]. Cyclic lipopeptides reduce the interfacial surface tension, thus enhancing bacteria’s motility and colonization, displaying subsequently potent antimicrobial activity [84,85,86]. Specifically, bacteria of the *B. amyloliquefaciens, B. velezensis*, *B. halotolerans*, *B. subtilis* and *B. mojavensis* species secrete a variety of surface-active antimicrobial compounds to control plant pathogens most commonly belonging to the iturin, surfactin and fengycin families [51,87,88]. The selected *Bacillus* strains, Cal.r.29, Cal.l.30, Cal.f.4, Cal.r.11, Cal.f.2.1, Cal.l.11 and Cal.r.19, belonging to the above species secreted surfactant compounds in both liquid and solid growth culture maintaining their strong antimicrobial activity against several plant pathogenic fungi tested under in vitro and ex vivo (detached tomato fruits) trails. Ethyl-acetate extracts of agar-diffusible secreted metabolites of singly grown or during confrontation with *B. cinerea*, produced strong bioactive compounds as determined in situ by the TLC- bioautography method. Secreted metabolites produced by each bacterial strain suppressed *B. cinerea* spore formation and mycelial growth forming distinct inhibition spots following the same pattern, indicating the ability of bacteria to produce constitutively the bioactive compounds. The BCA candidates presented more than one inhibition spots, corresponding most probably to homologues of iturinic lipopeptides (iturin A, bacillomycin D, mojavensin A) and/or surfactin homologues, with *B. halotolerans* strains presenting the same TLC profile depending on their R*f* values [49,89,90,91]. These lipopeptides are known to affect both the physiology and morphology of phytopathogenic fungi by forming pores in the cell membrane, creating swollen and deformed hyphae, increasing hyphal branching and disrupting spore membrane [86,92].

Inoculation of the best-performing endophytes to detached tomato fruits against the post-harvest pathogen *B. cinerea* led to great reduction of rot symptoms, as evident from the reduced disease severity index and disease incidence data. The least infected wounds were recorded in tomato fruits treated with *B. halotolerans* strains Cal.l.30 and Cal.f.4 and *B. velezensis* Cal.r.29, all great producers of antifungal compounds. The suppression of *B. cinerea* on detached tomato fruits was probably a combination of direct and indirect modes of action. All strains successfully colonized the point of application, competing directly with the pathogen for nutrients and space, secreted several antimicrobial secondary metabolites and lytic enzymes, and competed indirectly by inducing tomato fruits’ defense mechanisms through various ISR elicitors produced (e.g., secondary antimicrobial metabolites, lytic enzymes, phytohormones, BVCs and siderophores) [93,94,95,96]. According to the literature, the antimicrobial compounds produced are not only limited to their lytic action, but also contribute indirectly to plant protection by stimulating the plant’s induced systemic resistance (ISR), both in the contact tissue and systemically [97,98,99].

Our data indicated that *C. officinalis* collected from an environment with high drought and temperature hosts a wealth of beneficial endophytic bacteria belonging mostly to *B. halotolerans*, *B. velezensis* and *B. subtilis* species. Furthermore, considering together the multiple plant growth-promoting features, survivability in extreme growth conditions and effective biocontrol ability, *B. halotolerans* endophytic bacteria are posing as the most promising BCA candidates and plant stimulators. Formulation of bacteria as microbiota rather than as single strains may play a more important role in promoting host growth and combating biotic and abiotic stress [100]. Thus, our future goal is to test whether single or two-member *B. halotolerans* strain mixtures could be employed as plant growth biostimulants, systemic resistance inducers and biocontrol agents in horticultural crops under greenhouse conditions.

## Figures and Tables

**Figure 1 microorganisms-11-00206-f001:**
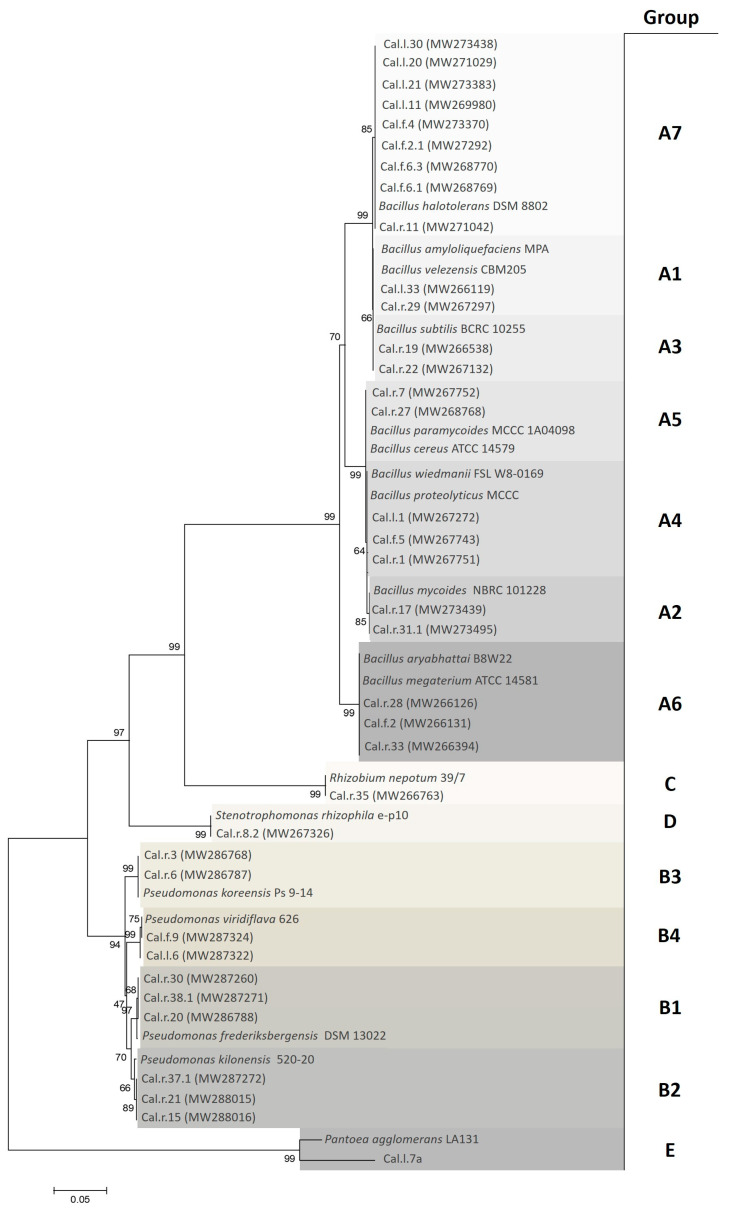
Phylogenetic tree based on 16S rRNA nucleotide sequences of endophytic bacterial isolates from *C. officinalis*. Bacterial endophytes are categorized in 14 colored groups (A1-7, B1-4, C, D and E) based on their similarity. The tree was constructed using the Neighbor-joining method with 1000 bootstrap data sets.

**Figure 2 microorganisms-11-00206-f002:**
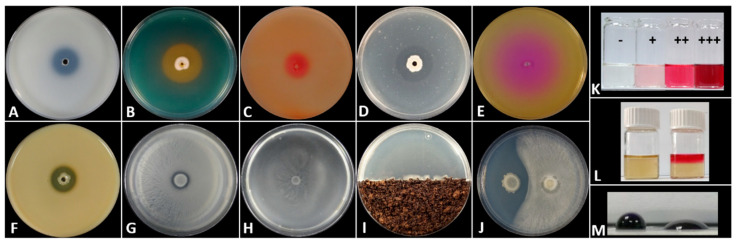
PGP traits, motility and biocontrol ability in vitro of bacterial endophytes of *C. officinalis*. (**A**) Phosphate solubilization (clear halo), (**B**) Siderophore production (yellow-orange halo), (**C**) Cellulase production (red halo), (**D**) Chitinase production (clear halo), (**E**) Urease production (pink halo), (**F**) Protease production (clear halo), (**G**) Swarming motility, (**H**) Swimming motility, (**I**) Chemotaxis, (**J**) Biocontrol of *F. oxysporum* (inhibition zone), (**K**) Indole-3-acetic acid (IAA) production (−: negative control, +: low production, ++: medium production, +++: high production), (**L**) Acetoin production (left: negative control, right: positive reaction) and (**M**) Drop collapse assay (left: negative control, right: positive reaction).

**Figure 3 microorganisms-11-00206-f003:**
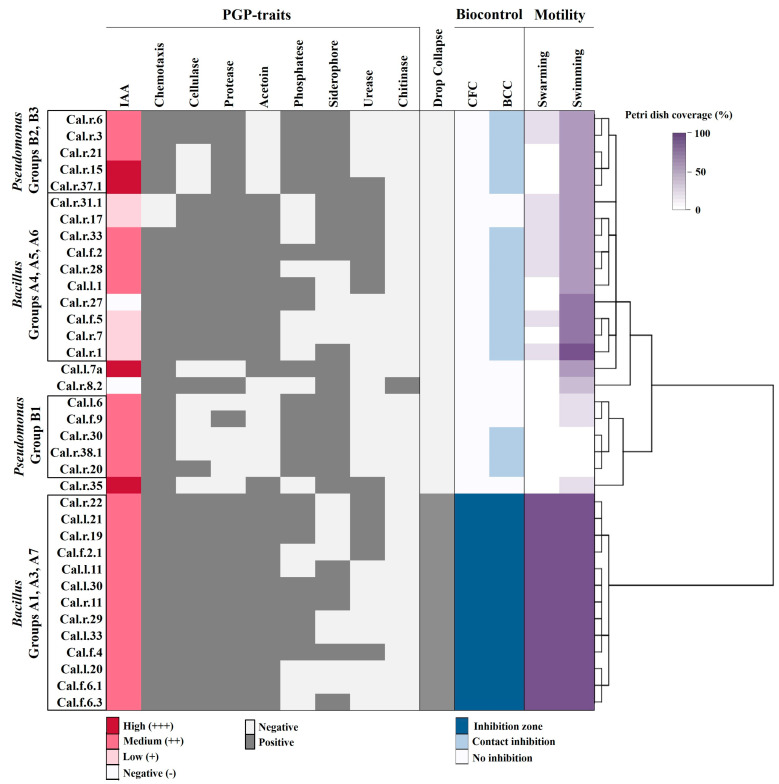
Hierarchically clustered heatmap of plant growth promoting (PGP) traits (light grey, positive; dark grey, negative), IAA production (white: negative, light pink: low production, pink: medium production, dark pink: high production), motility on different surfaces (% Petri dish coverage) and biocontrol capacity (white: no inhibition, light blue: contact inhibition, dark blue: inhibition zone) of BCC (Bacterial Cell Culture) and CFC (Cell-Free Culture). The hierarchical clustering was created to group the bacterial strains by similar reaction to the different assays tested.

**Figure 4 microorganisms-11-00206-f004:**
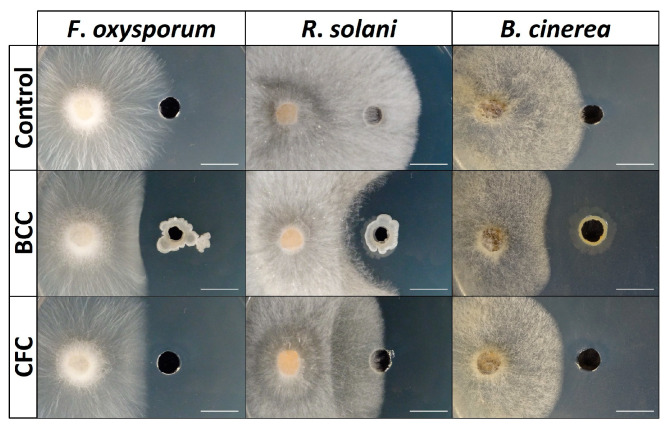
Dual culture assay for mycelial inhibition of the phytopathogenic fungi *F. oxysporum*, *R. solani* and *B. cinerea* by bacterial cell culture (BCC) and bacterial cell-free culture (CFC) with the creation of an inhibition zone (scale bar = 5 mm).

**Figure 5 microorganisms-11-00206-f005:**
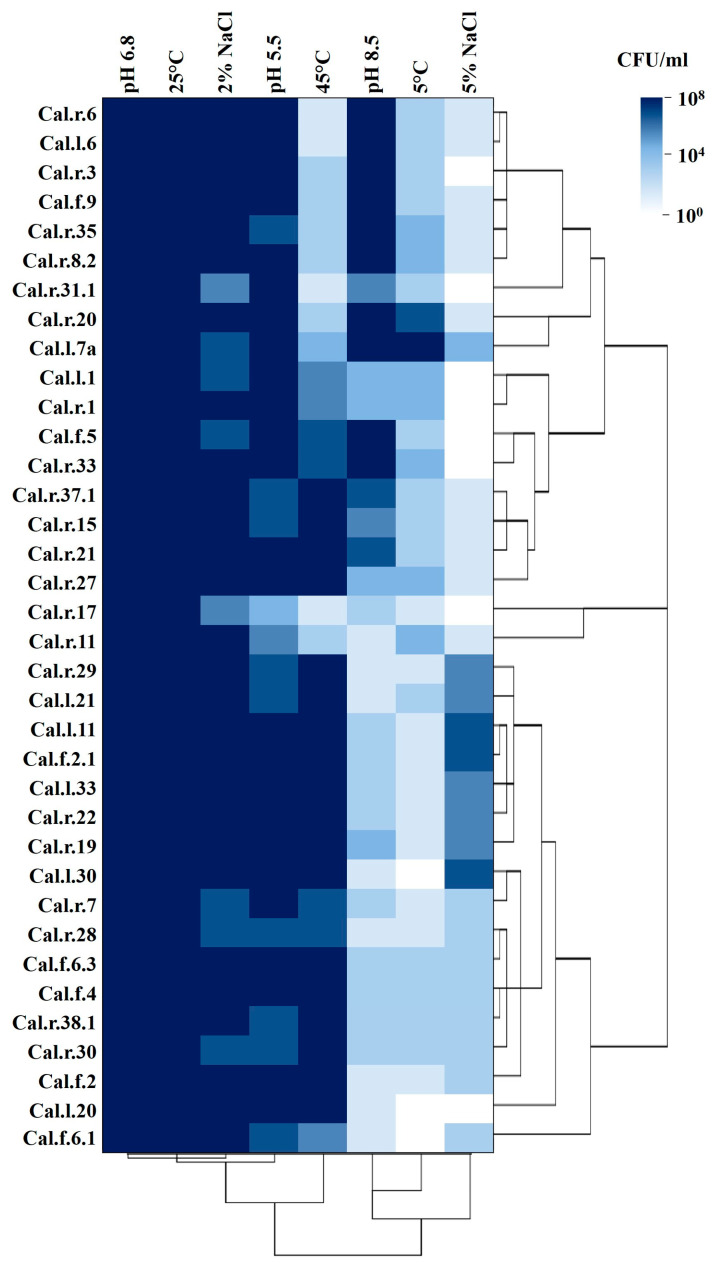
Survival in different growth conditions. The bacterial cell density is visualized using a hierarchically clustered heatmap. Different colors in the heatmap indicate different survival rates expressed in CFU/mL.

**Figure 6 microorganisms-11-00206-f006:**
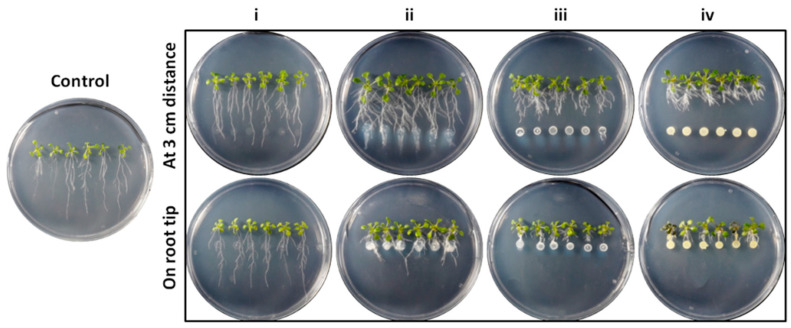
Co-cultivation of *A. thaliana* seedlings with endophytic bacterial strains inoculated on root tips and at 3 cm distance from root tips. Four morphological categories emerged after at-distance formulation: (**i**) plants with long PRL and decreased LRN, (**ii**) plants with long PRL and increased LRN, (**iii**) plants with intermediate PRL and increased LRN and (**iv**) plants with short PRL and increased LRN.

**Figure 7 microorganisms-11-00206-f007:**
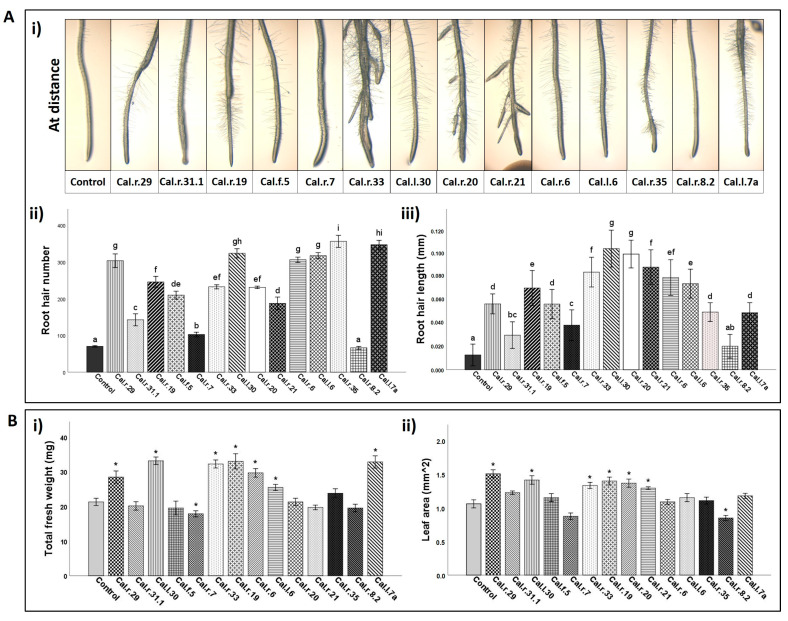
(**A**) Root morphology of *A. thaliana* seedlings after endophytic bacterial inoculation at-distance from plant root tips. (**i**) Microscopically observed roots; (**ii**) Root hair number (RHN) and (**iii**) root hair length (RHL) after bacterial formulation in comparison to Control plants. Data represent mean values (SD) of 12 seedlings and letters indicate statistically significant differences among treatments, based on Tukey’s test at *p* = 0.05. (**B**) Co-cultivation of *A. thaliana* seedlings with endophytic bacterial strains in two compartment Petri dishes (I-plates). (**i**) Total fresh weight (mg) and (**ii**) leaf area (mm^2^) of *A. thaliana* seedlings after bacterial inoculation, in comparison to Control plants. Data values represent the mean of 12 seedlings ± SD per treatment. Asterisks indicate the statistical difference between control plants and treated with bacteria plants after Dunnett’s test.

**Figure 8 microorganisms-11-00206-f008:**
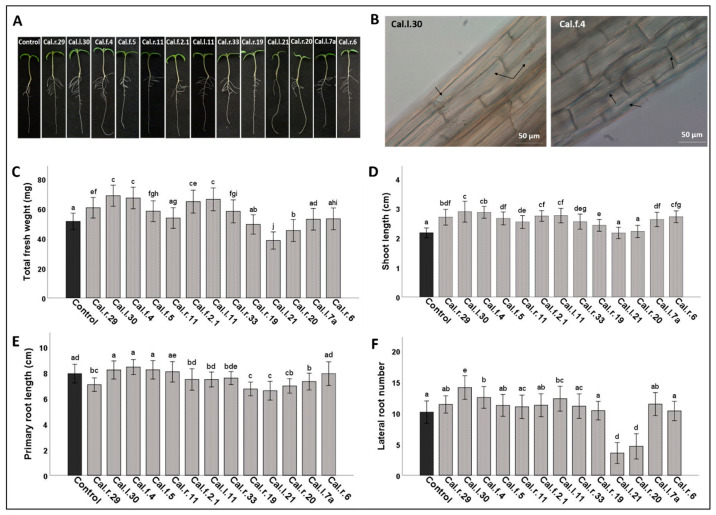
Tomato seed bio-priming with endophytic bacterial strains. (**A**) Tomato seedlings’ morphology, (**B**) Root colonization by endophytic strains Cal.l.30 and Cal.f.4., (**C**) Total fresh weight (mg), (**D**) Shoot length (cm), (**E**) Primary root length (cm) and (**F**) Lateral root number. Data values represent the mean of 45 seedlings ± SD per treatment. Different letters indicate statistically significant differences among treatments, based on Tukey’s test at *p* = 0.05.

**Figure 9 microorganisms-11-00206-f009:**
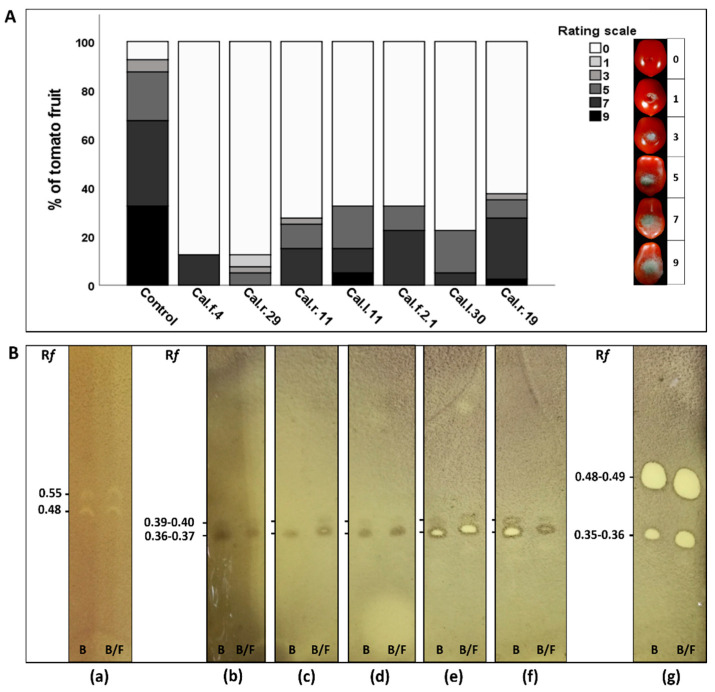
(**A**) Infected tomato fruit (%) as assessed by color in control (*B. cinerea*) and inoculated (both competitive bacterial strains and the pathogen) fruits (n = 20). Colored blocks within each column represent the percentage of fruit corresponding to the scale value of the disease severity of *B. cinerea*. Visual disease rating scale of symptoms caused by *B. cinerea* is represented as 0 = healthy fruits, 1 = 1–10%, 3 = 11–25%, 5 = 26–50%, 7 = 51–75% and 9 = >75% infected fruit area. (**B**) TLC-bioautography of antimicrobial agar diffusible secreted metabolites of bacterial endophytes (**a**) Cal.r.19, (**b**) Cal.l.11, (**c**) Cal.r.11, (**d**) Cal.f.2.1, (**e**) Cal.f.4, (**f**) Cal.l.30 and (**g**) Cal.r.29 produced when grown singly (**B**) and during confrontation with *B. cinerea* (B/F). Retention factor (R*f*) values are marked on the image next to each inhibition zone of *B. cinerea* mycelial growth.

**Table 1 microorganisms-11-00206-t001:** Plant growth effect of representative endophytic bacterial strains of each bacterial species on different growth parameters of *A. thaliana* seedlings under in vitro formulation. Data values represent the mean of 12 seedlings ± SD per treatment. Asterisks indicate the statistical difference between control plants and plants after bacterial formulation.

Group	Bacterial Species	Strain	AMG	Total Fresh Weight (mg)		Primary Root Length (cm)		Lateral Root Number (N)	
				AD ^A^	ORT ^B^	AD ^A^	ORT ^B^	AD ^A^	ORT ^B^
-	-	-	Control	10.2 ± 1.29	10.3 ± 2.19	5.8 ± 0.49	5.05 ± 0.16	8.75 ± 1.55	8.75 ± 1.55
A1	*B. velezensis*	Cal.r.29	ii	25.00 ± 1.02 *	18.65 ± 1.50 *	3.55 ± 0.25 *	1.78 ± 0.24 *	21.08 ± 2.35 *	12.17 ± 2.48 *
A2	*B. mycoides*	Cal.r.31.1	i	12.98 ± 0.73	17.53 ± 1.49.*	5.03 ± 0.39 *	4.24 ± 0.14 *	8.67 ± 1.61	10.58 ± 1.78
A3	*B. subtilis*	Cal.r.19	ii	27.40 ± 0.76 *	15.15 ± 1.12 *	3.31 ± 0.22 *	2.17 ± 0.20 *	18.5 ± 2.47 *	11.50 ± 2.02 *
A4	*B. proteolyticus*	Cal.f.5	i	13.40 ± 0.46 *	17.88 ± 1.20 *	4.23 ± 0.37 *	5.07 ± 0.25	12.91 ± 1.72 *	7.50 ± 1.57
A5	*B. cereus*	Cal.r.7	i	12.63 ± 1.66	13.72 ± 1.62 *	4.91 ± 0.37 *	5.63 ± 0.24	12.33 ± 1.61 *	10.00 ± 2.41
A6	*B. megaterium*	Cal.r.33	iii	13.73 ± 0.54 *	12.72 ± 1.07 *	2.31 ± 0.29 *	1.13 ± 0.09 *	14.83 ± 2.17 *	10.25 ± 1.06
A7	*B. halotolerans*	Cal.l.30	ii	31.92 ± 1.27 *	17.78 ± 1.38 *	3.34 ± 0.24 *	3.56 ± 0.29 *	17.25 ± 1.66 *	12.42 ± 3.58 *
B1	*Pseudomonas* sp.	Cal.r.20	iii	24.03 ± 2.24 *	13.05 ± 1.59 *	2.51 ± 0.45 *	1.46 ± 0.81 *	20.92 ± 4.44 *	14.33 ± 1.61 *
B2	*P. kilonensis*	Cal.r.21	iii	23.25 ± 1.66 *	12.60 ± 0.88	2.79 ± 0.37 *	1.64 ± 0.10 *	23.17 ± 1.95 *	23.42 ± 2.35 *
B3	*P. koreensis*	Cal.r.6	iii	22.25 ± 2.02 *	16.23 ± 1.49 *	2.71 ± 0.29 *	1.39 ± 0.09 *	22.42 ± 2.02 *	13.42 ± 1.08 *
B4	*P. viridiflava*	Cal.l.6	iii	17.32 ± 2.82 *	18.15 ± 1.34 *	3.08 ± 0.42 *	2.13 ± 0.18 *	27.17 ± 1.64 *	21.58 ± 2.54 *
C	*Rhizobium* sp.	Cal.r.35	iv	23.18 ± 1.66 *	11.32 ± 1.15	2.08 ± 0.15 *	1.15 ± 0.10 *	26.92 ± 1.93 *	9.67 ± 0.78
D	*Stenotrophomonas* sp.	Cal.r.8.2	i	13.33 ± 2.97	13.38 ± 0.85 *	5.73 ± 0.44	4.32 ± 0.24 *	11.67 ± 2.43 *	11.33 ± 2.77 *
E	*Pantoea* sp.	Cal.l.7a	iv	20.68 ± 2.49 *	10.82 ± 1.68	1.87 ± 0.23 *	1.04 ± 0.17 *	29.5 ± 2.51 *	9.08 ± 1.44

AMG: *A. thaliana* morphological group, ^A^: At 3 cm distance (AD), ^B^: On root tip (ORT), *: Significant differences with control plants (*p <* 0.05).

**Table 2 microorganisms-11-00206-t002:** Tomato seed germination after bacterial seed bio-priming. The values represent the average of the measurements and the standard deviations of the values. The asterisks indicate statistically significant differences (*p* < 0.05) between the values (n = 50) compared to the control plants per observation days after Dunnett’s test.

Bacterial Strains	% Tomato Seed Germination	
	3 dps	8 dps
Control	83.00 ± 1.00	10.3 ± 2.19
Cal.r.29	88.00 ± 3.00	18.65 ± 1.50 *
Cal.l.30	91.67 ± 2.52 *	17.53 ± 1.49 *
Cal.f.4	90.00 ± 1.00 *	15.15 ± 1.12 *
Cal.f.5	85.00 ± 1.73	17.88 ± 1.20 *
Cal.r.11	82.67 ± 3.06	13.72 ± 1.62 *
Cal.f.2.1	78.00 ± 3.00	12.72 ± 1.07 *
Cal.l.11	91.67 ± 1.53 *	17.78 ± 1.38 *
Cal.r.33	90.33 ± 1.53 *	13.05 ± 1.59 *
Cal.r.19	89.00 ± 2.65 *	12.60 ± 0.88
Cal.l.21	75.67 ± 2.52 *	16.23 ± 1.49 *
Cal.r.20	72.33 ± 2.52 *	18.15 ± 1.34 *
Cal.l.7a	85.67 ± 3.22	11.32 ± 1.15
Cal.r.6	81.00 ± 2.00	13.38 ± 0.85 *

**Table 3 microorganisms-11-00206-t003:** Biological control effect of selected bacterial endophytes against *Botrytis cinerea* on detached tomato fruit. Values represent the mean of three independent replicates and their standard deviations. Different letters indicate statistically significant differences among treatments, based on Tukey’s test at *p* = 0.05.

Treatment	Disease Severity Index (%)	Disease Incidence (%)
Control	72.52 ± 3.61a	91.67 ± 2.89a
Cal.f.4	12.85 ± 2.57b	13.33 ± 2.89b
Cal.r.29	11.49 ± 1.50b	15.00 ± 5.00b
Cal.r.11	23.66 ± 1.32cd	31.67 ± 7.64c
Cal.l.11	22.85 ± 0.66cd	36.67 ± 7.64c
Cal.f.2.1	29.28 ± 3.98d	35.00 ± 5.00c
Cal.l.30	17.49 ± 1.79bc	25.00 ± 5.00bc
Cal.r.19	30.53 ± 1.63d	38.33 ± 2.89c

## Data Availability

All 16S rRNA gene sequences’ accession numbers are available in the NCBI database. The *Bacillus halotolerans* strains Cal.l.30, Cal.f.4 and Cal.l.11 whole genome projects are available in the NCBI database under the accession numbers JAEACK000000000, JAEACI000000000 and JAEACJ000000000 (GenBank), respectively, and SAMN16949411, SAMN16949417, SAMN16949418 (BioSample) and PRJNA681331, PRJNA681333, PRJNA681332 (BioProject), respectively.

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
