# Peer review of "Calendula officinalis—A Great Source of Plant Growth Promoting Endophytic Bacteria (PGPEB) and Biological Control Agents (BCA)"

_microorganisms, 2023, doi:10.3390/microorganisms11010206_

Round 1

Reviewer 1 Report

Consider as selection criteria for sequencing the strains where the characteristics that promote plant growth were detected?

The document is interesting and presents advances in microbial diversity with characteristics of biotechnological interest.

Reviewer 2 Report

The Authors present a clean piece of work on the endophitic bacteria of Calendula officinalis. They extract the culturable bacteria, identify them and characterize them according to the most common PGPR traits. The work is well presented and the results are interesting.

My only concern is that it is a description, a report of what was done. There is no hypotheses to test. So this is my main suggestion is that the Authors present the working hypothesis.

When testing the effect of the isolates on plant traits I think it would be interesting to have a treatment with all the isolates, as all were isolated from the same plant.

Reviewer 3 Report

The manuscript addressed isolation, characterization, identification and biostimulating properties of the plant growth promoting endophytic bacteria obtained fromdifferent parts of the medicinal plant Calendula officinalis. 

The following points should be resolved by the authors to improve the manuscript before being considered as accepted.

1- The rationale of the work should be clearly highlighted.

2- Previous work reported in the literature about similar bacteria recoved from the same plant should be discussed throughly in the introduction. The gap that the current work needs to bridge should be adressed.

3- The source of the fungal strains, plant seeds ,..etc should  be mentioned.

4- The aim of certain experiments should be indicated. 

5- there is an issue in the figure numbers and in the text. Figures' legands should be self-explanatory, details about the observations should be mentioned clearly. 

7- The conclusion/recommendation and future work should receive a much more stress to value this study.

8- Scientific names should be italic in the entire manuscript. 

9- Use scientific names'abbreviations properly throughout the manuscript. (plant and bacterial names as well as the media)

10- References should be reduced in number (you can remove the repeated old ones and keep the most relevant and latest ones, when possible).

The entire manuscript should be revised by an English native speater. 

Details are also included in the edited file attached. 
